# Large-area and efficient perovskite light-emitting diodes via low-temperature blade-coating

Shenglong Chu [1,3], Wenjing Chen[1,3], Zhibin Fang[1,3], Xun Xiao [2], Yan Liu[1], Jia Chen[1], Jinsong Huang [2] & Zhengguo Xiao [1✉]

Large-area light-emitting diodes (LEDs) fabricated by mass-production techniques are needed for low-cost flat-panel lighting. Nevertheless, it is still challenging to fabricate efficient large-area LEDs using organic small molecules (OLEDs), quantum dots (QLEDs), polymers (PLEDs), and recently-developed hybrid perovskites (PeLEDs) due to difficulties controlling film uniformity. To that end, we report sol-gel engineering of low-temperature blade-coated methylammonium lead iodide ($MAPbI_3$) perovskite films. The precipitation, gelation, aging, and phase transformation stages are dramatically shortened by using a diluted, organoammonium-excessed precursor, resulting in ultra-flat large-area films ($54\,cm^2$) with roughness reaching 1 nm. The external quantum efficiency of doctor-bladed PeLEDs reaches 16.1%, higher than that of best-performing blade-coated OLEDs, QLEDs, and PLEDs. Furthermore, benefitting from the throughput of the blade-coating process and cheap materials, the expected cost of the emissive layer is projected to be as low as 0.02 cents per $cm^2$, emphasizing its application potential.

[1] Hefei National Laboratory for Physical Science at the Microscale, Department of Physics, CAS Key Laboratory of Strongly-Coupled Quantum Matter Physics, University of Science and Technology of China, Hefei, Anhui 230026, China. [2] Department of Applied Physical Sciences, University of North Carolina at Chapel Hill, Chapel Hill, NC 27599, USA. [3] These authors contributed equally: Shenglong Chu, Wenjing Chen, Zhibin Fang. ✉email: zhengguo@ustc.edu.cn

Solution-processed organic small molecule light-emitting diodes (OLEDs), quantum dot LEDs (QLEDs), and polymer LEDs (PLEDs) have been intensively studied for decades and have experienced great progress in small-area devices[1–3]. The external quantum efficiency (EQE) of solution-processed small-area OLEDs, QLEDs, and PLEDs have reached >20% using spin-coating[4–7]. Nevertheless, large-area fabrication of these LEDs using mass-production techniques such as blade-coating, spray-coating, and ink-jet printing, etc., is challenging because of the difficulty in control of film thickness, uniformity/coverage, crystallinity, etc., resulting in much lower EQE than the devices made by spin-coating[8–10], and limiting potential applications of those LEDs in flat panel lighting.

Metal halide perovskites (MHPs) have become a new generation of light-emitting materials due to their fantastic optoelectronic properties, such as high color purity, tunable band gaps, bipolar conduction, etc.[11–15]. The EQE of perovskite LEDs (PeLEDs) has been boosted to >20% with a small device area of <0.1 cm² (refs. [16–25]) and has recently reached 12.1% with an area of 9 cm² using spin-coating[26]. Nevertheless, perovskite films made by mass-fabrication techniques such as blade-coating are usually very rough and not continuous due to coffee ring effect and fast crystallization of perovskites[27]. Blade-coating at high temperatures above 100 °C, incorporation of additives/surfactant, or assistance of air knife have been developed to improve the morphology of thick perovskite films for solar cells[28,29]. However, the thick films with large grains are not suitable for PeLEDs due to their low electron–hole capture rates[20]. Therefore, the EQE of PeLEDs made by mass-production techniques reached only 1.1%, much lower than the devices fabricated by spin-coating[30].

In this work, we report highly efficient PeLEDs fabricated by a robust blade-coating approach at a low temperature of 50 °C. The sol–gel stages of the doctor-bladed methylammonium lead iodide ($CH_3NH_3PbI_3$, $MAPbI_3$) films are effectively modified by changing precursor concentration and incorporation of excess bulky organoammonium halides. The diluted precursor with excess organoammonium halides results in much denser nucleation centers, slower/eliminated gelation process, and faster phase transformation process. As a result, the roughness of perovskite films decreases dramatically from 394.7 to 8.7 nm with precursor concentration reducing from 0.8 to 0.02 M and further decreases to around 1 nm with 50% molar excess of bulky organoammonium halides. The large-area, blade-coated films show great uniformity in terms of thickness, roughness, and optoelectronic properties. After incorporating the uniform film as an emissive layer, the EQE of PeLEDs reaches 16.1% with an area of 0.04 cm² and 12.7% with an area of 1 cm². The PeLEDs with an ultra-large area of 28 cm² with uniform emission is also demonstrated. Furthermore, the cost of the emissive layer reaches 0.02 cents per cm², demonstrating its huge potential in real applications of flat panel lighting and display.

## Results

### Doctor-blading of ultra-thin uniform perovskite films.
Figure 1a shows a schematic of blade-coating process. The depth of applicator is fixed around 2–3 μm to ensure relatively thin perovskite films. The concentration of $MAPbI_3$ precursors varies broadly from 0.02 to 0.8 M in dimethylformamide (DMF). A $N_2$ knife is applied to accelerate solvent evaporation and improve film morphology. No antisolvent is used in the blade-coating process. It should be noted that the blade-coating speed, temperature, and the $N_2$ knife pressure affect the film morphology dramatically (Supplementary Figs. 1–3). An ultra-fast coating speed of 150 mm/s, corresponding to Landau–Levich mode that the as-coated layer is still wet right after blading[29], a low temperature of 50 °C, and a $N_2$ knife pressure of

0.2 MPa are adopted for the coating process. Details of the optimization process can be found in Supplementary Information. As shown in Fig. 1b, the thicknesses of the films linearly depend on the precursor concentration, resulting from the Landau–Levich coating mode, and are slightly thinner with $N_2$ knife. Incorporation of bulky organoammonium halides, 4-fluorophenylmethylammonium iodide (FPMAI), does not change the thickness obviously. The thicknesses of the blade-coated films are very repeatable with acceptable fluctuations (Supplementary Fig. 1).

Atomic force microscopy (AFM) and scanning electron microscopy (SEM) measurements are performed to examine surface morphology of the blade-coated films. It is unexpected that the perovskite grains become smaller and smaller, and the films become more and more uniform with precursor concentrations decreasing from 0.8 to 0.02 M (Fig. 1c and Supplementary Fig. 4a). Accordingly, the film roughness reduces dramatically from 394.7 nm (film thickness ~950 nm) to 8.7 nm (film thickness ~25 nm). The films prepared with the aid of $N_2$ knife are smoother with smaller grains due to the faster DMF evaporation speed and higher nucleus density (Fig. 1d and Supplementary Fig. 4b). The roughness reduces accordingly from 60.8 to 1.6 nm with precursor concentrations decreasing from 0.8 to 0.02 M. Nevertheless, the rod-like grains for the thick films suggest that $N_2$ knife will not affect grain growth mechanism.

Interestingly, after incorporation of 50% molar excess FPMAI, the perovskite grains change from rod-like to circle-like (Fig. 1e, Supplementary Fig. 5a), clearly indicating that the grain growth of perovskite is suppressed. No low-dimensional perovskite X-ray diffraction (XRD) peaks are observed in the film with FPMAI (Supplementary Fig. 7), indicating that FPMA ligands work as surfactant self-assembled at the three-dimensional perovskite nanocrystal surface. Notably, the film roughness is dramatically reduced from 131.8 to 4.3 nm with precursor concentration decreasing from 0.8 to 0.02 M. The roughness is further reduced to 0.8 nm using a 0.02 M precursor solution with $N_2$ knife (Fig. 1f and Supplementary Fig. 5b). A high-resolution SEM image shows that the grain sizes are around 10 nm (Supplementary Fig. 6), comparable to the optimized spin-coated films[20]. The roughness of the films prepared by different methods and precursor concentrations are plotted in Fig. 1b for comparison.

### Sol–gel stages of the film formation process.
In order to figure out the origin of improved film morphology using very dilute precursor solutions and with excess FPMAI, we examined the film-forming process using an optical microscope (Supplementary Movies 1–4). The snapshots of the videos are shown in Supplementary Fig. 8. We draw schematics of the film formation process using the sol–gel knowledge according to the videos (Fig. 2). Four stages of sol–gel process can be clearly observed from the videos for the dense precursor (0.8 M) without excess FPMAI or $N_2$ knife. Stage 1. $PbI_2$ precipitation. The as-coated wet film/ink-layer quickly becomes saturated for $PbI_2$ due to its relatively lower solubility than the organic component. Therefore, $PbI_2$ precipitate first in the form of $PbI_2$-DMF-MAI complex[31]. This stage starts at around 4.3 s after blade-coating with nucleus density around $1.43 \times 10^2/mm^2$ (Supplementary Fig. 8a). Stage 2. Gelation. The $PbI_2$-DMF-MAI nucleus grows quickly because of the saturated $PbI_2$ wet film/ink-layer. Stage 3. Aging. With DMF continuing to evaporate, the $PbI_2$-DMF-MAI complex continues growing and small $PbI_2$-DMF-MAI complex grains start to precipitate. In addition, the $PbI_2$-DMF-MAI complex starts to convert to perovskite phase as the concentration of organic component starts increasing. Stage 4. Phase transformation. As most DMF evaporate out, $PbI_2$ continues reacting with MAI and forms perovskite. This step finishes at around 14.7 s.

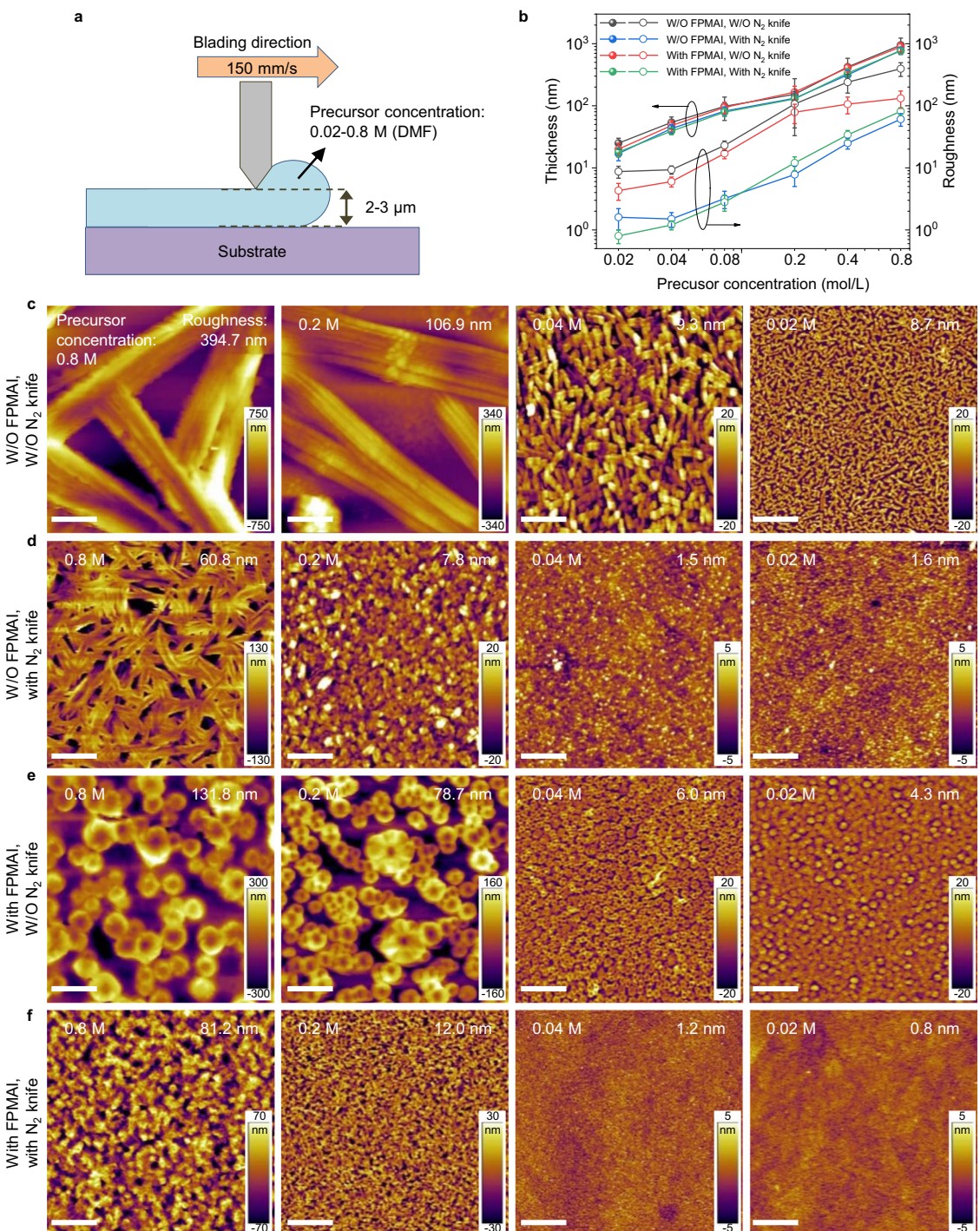

**Fig. 1 Optimization of doctor-bladed perovskite films. a** A schematic of the doctor-blading process. **b** Thickness and roughness as a function of precursor concentration for different fabrication methods. The statistics of thickness and roughness of the films are based on 15 scans (thickness profiler) and 3 scans (AFM), respectively. Error bars represent standard deviation. **c-f** AFM surface morphologies of the blade-coated films fabricated W/O FPMAI or $N_2$ knife (**c**), W/O FPMAI and with $N_2$ knife (**d**), with FPMAI and W/O $N_2$ knife (**e**), and with FPMAI and $N_2$ knife (**f**). The precursor concentrations and film roughness are also marked in the AFM images. The scale bars are 5 μm.

Excitingly, the sol–gel stages change substantially for the dilute precursor (0.2 M) (Supplementary Movie 2). The schematics of the film-forming process are shown in Fig. 2b. Stage 1. $PbI_2$ precipitation. With DMF evaporating, $PbI_2$ precipitates near the surface. But $PbI_2$ precipitations do not grow as the wet film/ink-layer is not $PbI_2$ saturated. This stage starts at around 4.4 s. Stage 2. More $PbI_2$ precipitation and gelation. With DMF continuing

evaporating, more $PbI_2$ precipitates near the surface and the existing $PbI_2$ precipitates grow slowly (Fig. 2b and Supplementary Fig. 8b). As a result, the nucleus density reaches $1.85 \times 10^4/mm^2$, which is two orders magnitude higher than the case using dense precursor (Supplementary Fig. 8b). Stage 3. Aging. The $PbI_2$-DMF-MAI complexes continue growing slowly and start to convert to perovskite phase. Stage 4. Phase transformation.

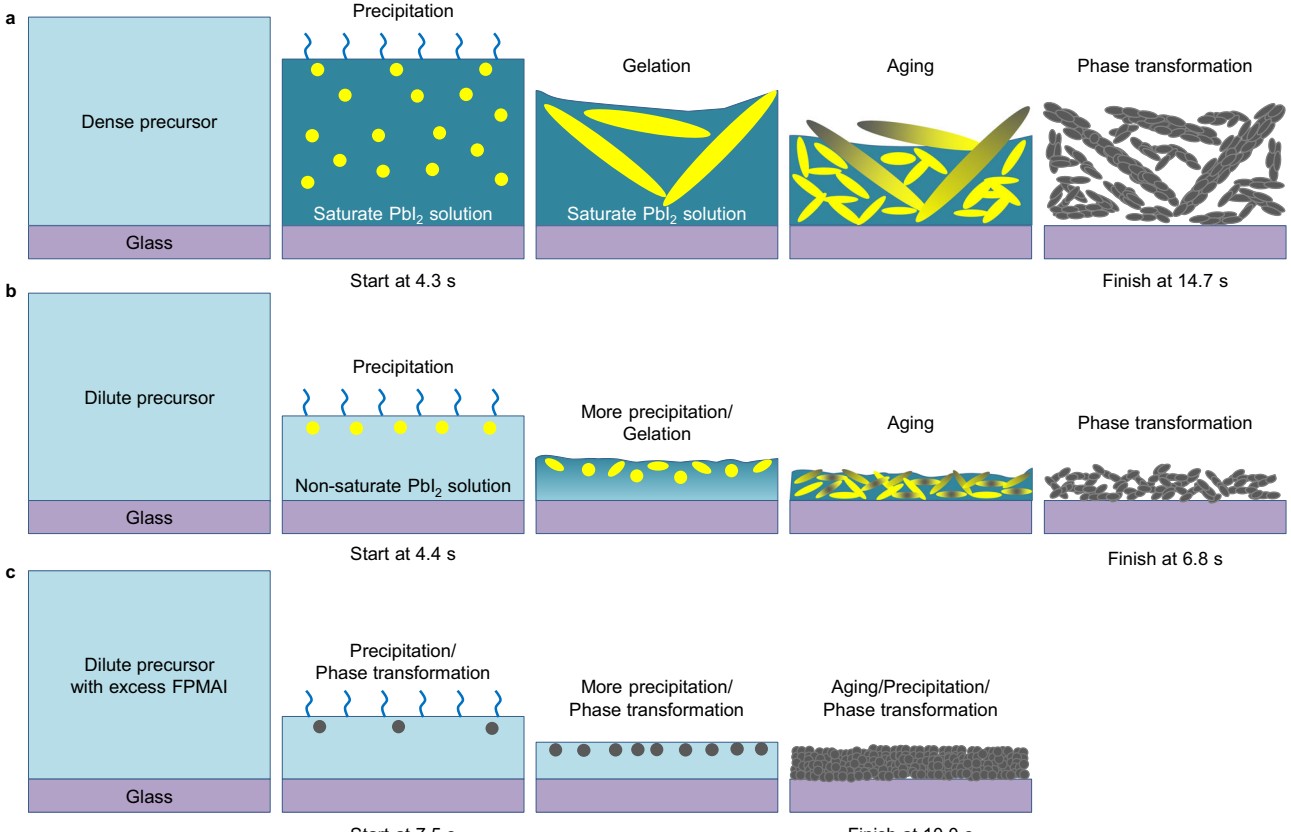

**Fig. 2 Schematics of sol–gel stages for different blade-coating methods. a–c** Schematics of the sol–gel stages using a dense precursor solution (0.8 M) (**a**), a dilute precursor solution (0.2 M) (**b**), and a dilute precursor solution with 50% molar excess FPMAI (**c**). All schematics are drawn according to the in situ optical microscopic images and the videos (Supplementary Fig. 5 and Movies) of the film formation processes without $N_2$ knife. The film formation process with $N_2$ knife or using dilute solutions with concentration <0.2 M are too fast to be resolved by optical microscope. The timing for precipitation beginning and phase transformation ending are also marked below the schematics.

Notably, the duration of this stage is very short because of the high nucleus density. This stage finishes at around 6.8 s.

More excitingly, the sol–gel stages are totally different after incorporation of excess FPMAI in the precursors (Supplementary Movies 3 and 4). The schematics of film formation process using a dilute precursor (0.2 M) are shown in Fig. 2c. Stage 1. $PbI_2$ precipitation/phase transformation. The $PbI_2$ precipitation and phase transformation process happen simultaneously at the surface, indicated by the black nucleus at the beginning (Supplementary Fig. 8c, d). This is because the excess organoammoniums in the precursor promotes reaction between $PbI_2$ and MAI/FPMAI. It also should be noted that the timing for $PbI_2$ precipitation is 7.5 s, in contrast to 4.4 s for the case without FPMAI. This is because excess FPMAI can increase the solubility of $PbI_2$. Stage 2. More $PbI_2$ precipitation and phase transformation. With DMF evaporating, more $PbI_2$ (or $MAPbI_3$ grain) precipitates near the surface. It should also be noted that the existing $MAPbI_3$ grains do not obviously grow in this stage because bulky FPMA ligands suppress grain growth. This results in a high nucleus density of $4.53 \times 10^5/mm^2$ (Supplementary Fig. 5a), more than one order magnitude higher than the case without FPMAI. Stage 3. Aging and phase transformation. As DMF fully evaporate out, the perovskite film with small grains forms. $N_2$ knife accelerates DMF evaporate speed and induce much denser nucleus. Therefore, the film formation processes are dramatically shortened and cannot be resolved by in situ optical microscope.

**Large-area perovskite films fabricated by doctor-blading.** To manifest the robustness of our blade-coating process, large films are prepared with an area of $6 \times 9$ cm$^2$ using the 0.04 M precursor solution. As a comparison, we also prepared films with the same size and thickness using a solvent-exchange approach, one of the most reliable methods to prepare high-quality uniform films. Notably, a higher precursor concentration of 0.2 M should be used to fabricate films with similar thickness with blade-coating (~40 nm). Furthermore, only a small volume of precursor (30 µl) is needed for blade-coating, while around 300 µl is required for spin-coating to ensure that the precursor can spread over the whole substrate. The applicator depth is critical to fabricate large-area uniform films. The 10 µm applicator results in precursor flowing on the substrate and non-uniform film (Supplementary Fig. 9). As shown in Fig. 3a, b, the blade-coated film is very uniform, while the spin-coated film shows circle-like morphology, possibly resulting from different solvent evaporation speed between the center and edge area during the spinning process.

The macroscale and microscale morphology uniformity of the large-area films are examined by thickness profiler and AFM, respectively. The large-area films were cut into small pieces with $1 \times 1$ cm$^2$ in size and marked as A$_{xy}$ ($x$, $y = 1, 2, 3…$) for the convenience of characterizations (Fig. 3c). Notably, the thickness of the bladed-coated film is very uniform over the whole substrate, while the spin-coated film is thinner at the center and thicker at the edge area (Fig. 3d, e). Excitingly, the blade-coated film is extremely uniform even at the microscale over the

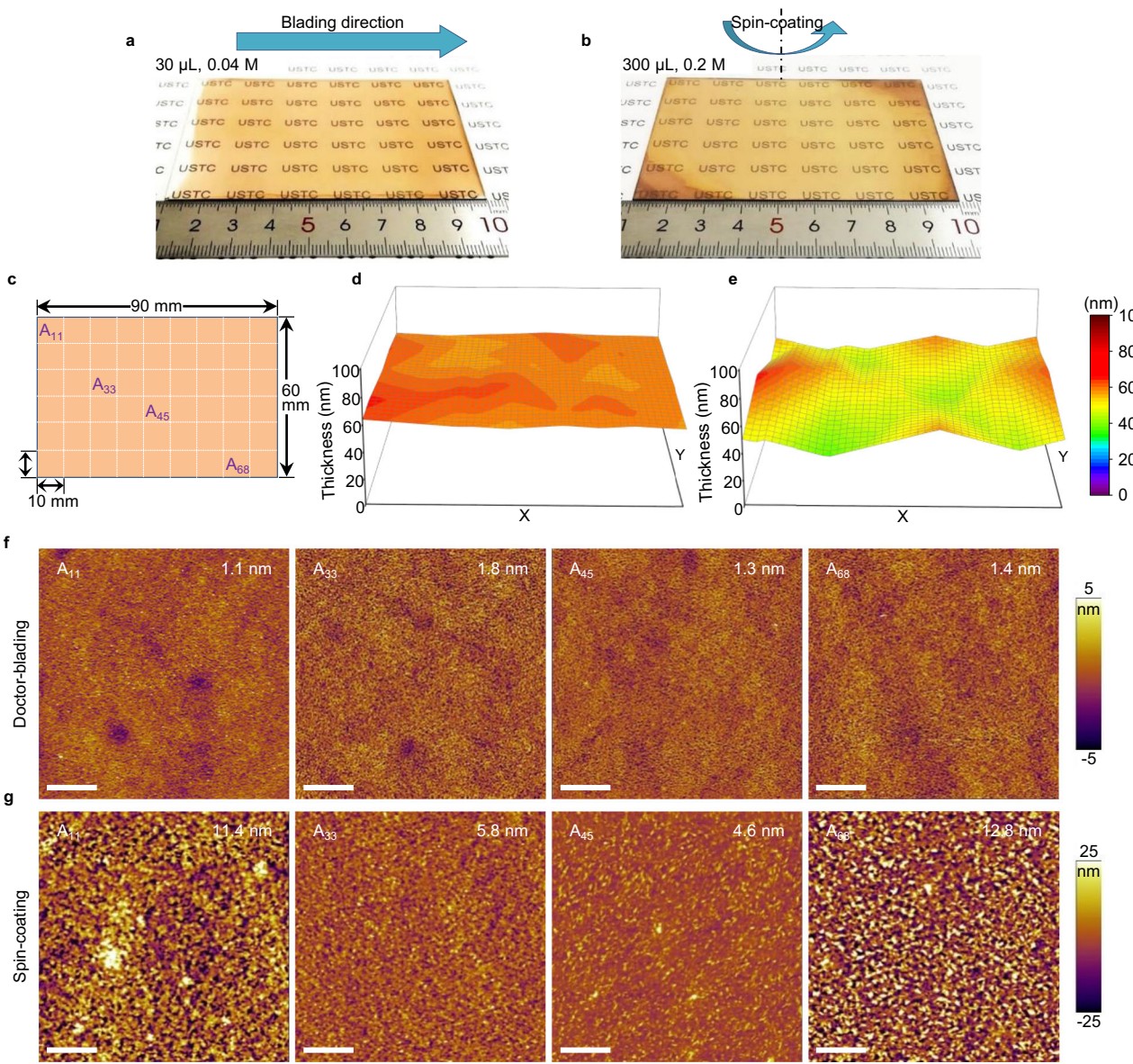

**Fig. 3 Morphology characterizations of large-area perovskite films. a, b** Photo images of the large-area perovskite films fabricated by doctor-blading (**a**) and spin-coating (**b**). The film area is 6 × 9 cm². The volume and concentration of the precursor solution needed to prepare the films are also marked in the images. **c** Schematics of the large-area film divided to 1 × 1 cm² pieces. Each pieces are marked as $A_{xy}$ (x, y = 1, 2, 3 ...). **d, e** Thickness mapping of the large-area films fabricated by doctor-blading (**d**) and spin-coating (**e**). The thicknesses of each small-area films (56 films in total) are measured separately. The three-dimensional figures are plotted by spatial interpolation. **f, g** AFM images of the large-area films fabricated by doctor-blading (**f**) and spin-coating (**g**). The number of small-area films and roughness are marked in the images. All scale bars are 5 μm.

whole substrate with roughness around 1 nm (Fig. 3f). In contrast, the roughness of the spin-coated film reaches 4–5 nm at the center and >10 nm at the edge area (Fig. 3g).

We then examine the uniformity of optoelectronic properties of the large-area films. Figure 4a, b show photoluminescence (PL) images of blade-coated and spin-coated films under an ultraviolet lamp. The PL emission of the blade-coated film is very uniform, agreeing well with its uniform morphology. In contrast, the PL emission is weaker in the edge area for the spin-coated film. This should be due to its rough morphology and/or insufficient DMF extraction during the solvent-exchange process. We then measured PL quantum yield (PLQY) and time-resolved PL (TRPL) of the films. The steady-state PL spectra and TRPL decay curves are shown in Supplementary Fig. 10. Excitingly, the blade-coated film is very uniform in steady-state PL peak positions and

intensities, as well as in TRPL decay. In contrast, the spin-coated films show relatively large fluctuations. The PLQY and average carrier lifetime ($T_{average}$) mapping of the large-area films are shown in Fig. 4c–f. The blade-coated film shows higher PLQY and longer $T_{average}$ than the spin-coated film with better uniformity. The result of microscale PL intensity mapping at the center area of the large films also demonstrates that the blade-coated film has stronger PL emission than the spin-coated film (Supplementary Fig. 11). Figure 4g, h and Supplementary Tables 1 and 2 show radiative and non-radiative recombination rates calculated using equation $\frac{1}{T_{average}} = k_{rad} + k_{nonrad}$ and $PLQY = \frac{k_{rad}}{k_{rad} + k_{nonrad}}$. The blade-coated film shows relatively lower non-radiative recombination and higher radiative recombination than the spin-coated film, which is critical for light-emitting devices.

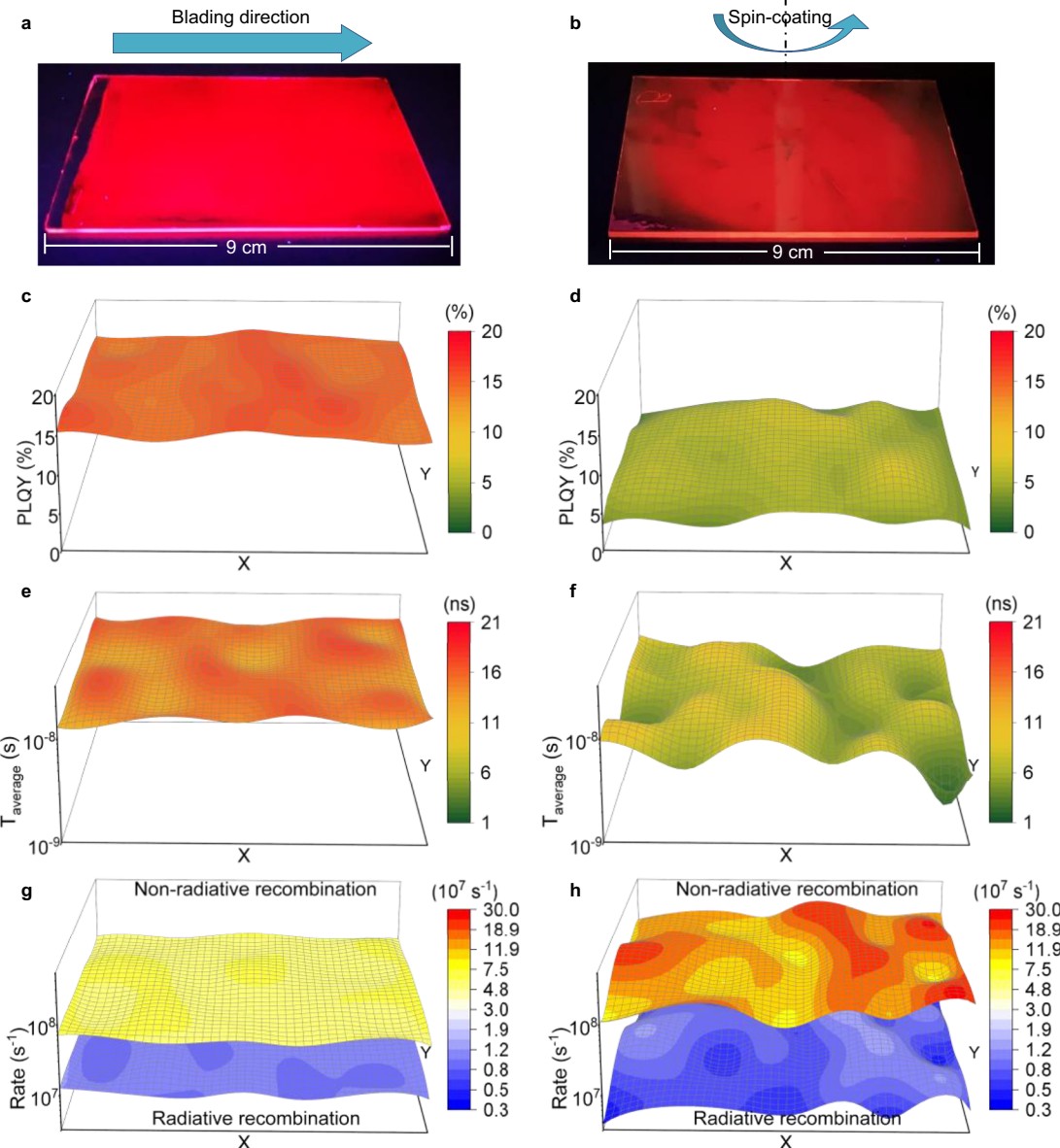

**Fig. 4 Optoelectronic characterizations of large-area perovskite films. a, b** PL images of the large-area perovskite films fabricated by doctor-blading (**a**) and spin-coating (**b**). The film area is 6 × 9 cm². **c–h** PLQY (**c, d**), $T_{average}$ (**e, f**), and radiative and non-radiative recombination rate (**g, h**) mapping of large-area films fabricated by doctor-blading (**c, e, g**) and spin-coating (**d, f, h**). The excitation wavelength and intensity for PLQY (TRPL) are 430 nm (369 nm) and 8 mW/cm² (6 mW/cm²), respectively. The characterizations are performed on small-area films (56 films in total). The three-dimensional figures are plotted by spatial interpolation.

**Efficient PeLEDs fabricated by doctor-blading**. PeLEDs are fabricated with a structure of indium tin oxide (ITO)/poly-TPD/perovskite/TPBi/LiF/Al (poly-TPD: poly[$N,N'$-bis(4-butylphenyl)-$N,N'$-bis(phenyl)-benzidine], TPBi: 2,2′,2″-(1,3,5-benzinetriyl)-tris(1-phenyl-1-H-benzimidazole), LiF: lithium fluoride). The device structure is shown in Fig. 5a. The molar ratio of FPMAI is fixed at 50% as it results in higher EQE than other ratios (Supplementary Fig. 12). As shown in Fig. 5b, all PeLEDs with FPMAI and N₂ knife have very low leaking current in the range of $10^{-5}$ mA/cm², resulting from the uniform and compact film. Notably, the highest EQE reaches 16.1% using 0.04 M precursor (Fig. 5c), higher than that of the state-of-the-art spin-coated devices based on MAPbI₃ (ref. [32]). The electro-luminescence (EL) spectra of the PeLEDs are shown in Fig. 5d. The blue shift of EL peaks for the ultra-thin perovskite films using

dilute precursors should result from dielectric confinement effect by the bulky ligands self-assembled at the perovskite nanocrystal surface and/or adjacent organic electron/hole transporting layers[33], considering that the grain sizes are larger than the Bohr radius[34]. The angular intensity profiles and EL spectra are shown in Supplementary Fig. 13. The EL spectra of PeLEDs with different thicknesses overlap for different angles. The EQE statistics of PeLEDs fabricated by different methods are shown in Fig. 5e and Supplementary Fig. 16. The devices without additive show relatively poor performances with EQE <1% (Supplementary Fig. 14a–d). This should be due to the rough films as well as low electron–hole binding energy of large MAPbI₃ grains. The EQE are generally higher with thinner films, resulting from more uniform film and better confinement of the electrons and holes. The PeLEDs with excess FPMAI show relatively high EQE in the

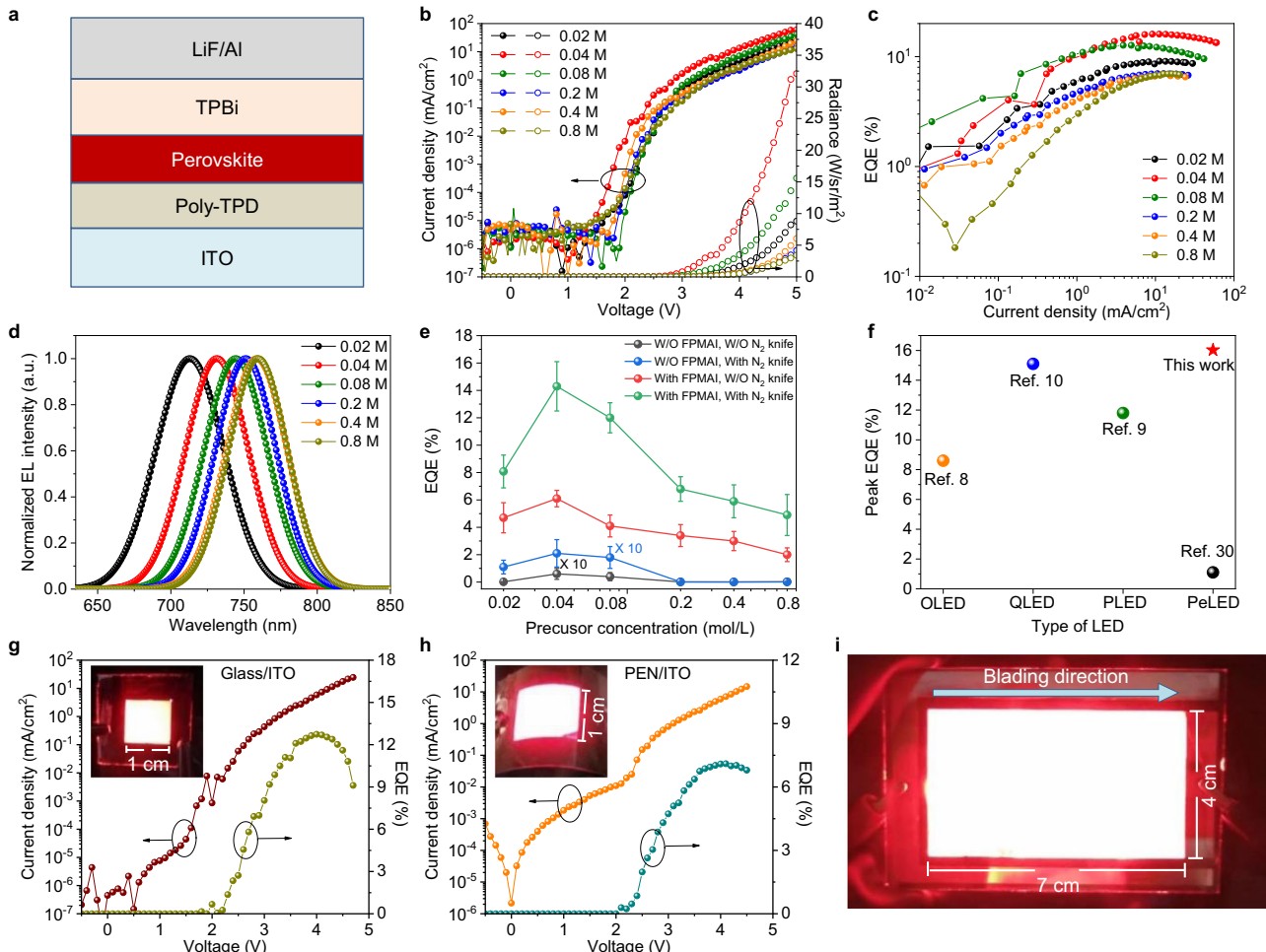

**Fig. 5 Performance of doctor-bladed PeLEDs. a** Device structure of PeLEDs. **b–d** Current density–voltage–radiance ($J$–$V$–$R$) curves (**b**), EQE (**c**), and EL spectra (**d**) of the PeLEDs fabricated with FPMAI additive and $N_2$ knife. **e** EQE statistics of the PeLEDs fabricated by different methods. The EQE statistics of PeLEDs made from 0.04 M solution with FPMAI additive and $N_2$ knife is based on 20 devices, and other categories are based on 10 devices. Error bars represent standard deviation. **f** Comparison of the peak EQE of OLEDs, QLEDs, PLEDs, and PeLEDs fabricated by blade-coating. **g, h** Current density and EQE curves of doctor-bladed PeLEDs on glass/ITO (**g**) and PEN/ITO (**h**) with a device area of $1 \times 1 \, cm^2$. Inset is photo image of the PeLED. **i** Photo image of an ultra large-area PeLED with a device area of $4 \times 7 \, cm^2$.

range of 1–10% even without $N_2$ knife for different perovskite thickness (Supplementary Fig. 14e, f), resulting from their relatively small grains.

Notably, the performance of our doctor-bladed PeLEDs has exceeded that of best-performing doctor-bladed OLEDs and PLEDs, which have been intensively studied for decades[8,9]. Figure 5f compares the peak reported EQEs of various types of LEDs fabricated by blade-coating. The superior performance of blade-coated PeLEDs should result from the extreme uniform blade-coated perovskite films fabricated by sophisticated control of sol–gel processes. The sol–gel engineering approach is expected to be applicable to other types of solution-processed LEDs, including OLEDs, QEDs, and PLEDs, to improve their performances as they are generally also sol–gel processes.

To further demonstrate the robustness of doctor-blading of PeLEDs, large-area devices are fabricated on both rigid glass/ITO substrates and flexible polyethylene naphthalate (PEN)/ITO substrates. The layout of our large-area devices is shown in Supplementary Fig. 15a. The poly-TPD layer is also fabricated by doctor-blading with a thickness of around 30 nm to ensure large-area uniformity. As shown in Supplementary Fig. 15b, the doctor-bladed poly-TPD layer is very flat with roughness around 0.2 nm, comparable to the spin-coated films. Notably, the devices show a

decent EQE of 12.7 and 7.1% on rigid and flexible substrates, respectively (Fig. 5e, f). The insets show photo images of the large-area PeLEDs with a device area of $1 \times 1 \, cm^2$ on glass/ITO substrates and flexible PET/ITO, respectively. An ultra-large PeLEDs with a device area of $4 \times 7 \, cm^2$ is also fabricated. As shown in Fig. 5i, the device has uniform EL emission over the whole working area, demonstrating the possibility of large-area production of PeLEDs. We finally measured the operational lifetime of the PeLEDs with different areas. As shown in Supplementary Fig. 17, the operational lifetime reached 19.6, 11.0, and 10.1 h for device area of 0.04, 1, and 28 $cm^2$, respectively, comparable to the state-of-the-art spin-coated devices[26]. Furthermore, the doctor-bladed PeLEDs also has superior advantages in terms of cost, which is critical for flat panel lighting applications. The cost of the emissive layer is calculated to be around 0.02 cents per $cm^2$, demonstrating huge potential applications of PeLEDs in flat panel lighting.

## Discussion
In conclusion, we have shown that the sol–gel stages of the blade-coating process of perovskite films can be effectively modified by changing precursor concentration and incorporation of excess

bulky organoammonium halides. The durations of the various sol–gel stages are dramatically shortened from over 10 s using dense precursor solutions to approximately 2.5 s using diluted precursor solutions, resulting from the increased density of nucleation centers. The sol–gel stages are further modified after incorporation of excess bulky FPMAI in the precursor. The $PbI_2$ precipitation and phase transformation to $MAPbI_3$ happen almost simultaneously because of the excess organoammonium content. In addition, no obvious gelation process is observed because bulky FPMAI suppresses grain growth. As a result, high-quality doctor-bladed films are prepared with great uniformity in film thickness and roughness, as well as optoelectronic properties. The EQE of PeLEDs incorporating these uniform films reached 16.1%, a world record value for blade-coated LEDs. Our results demonstrate the potential of mass fabrication of PeLEDs for flat panel lighting applications and others.

## Methods

**Materials.** MAI and FPMAI were synthesized by mixing methylamine (40 wt% in $H_2O$) and 4-fluorophenylmethylamine (Sigma Aldrich), respectively, with equimolar amounts of aqueous HI (Sigma Aldrich, stabilized) at 0 °C with constant stirring for 2 h. The solvent was removed using a rotary evaporator. The MAI and FPMAI were then recrystallized two times in $N_2$ glovebox. Recrystallization was performed by slow cooling their ethanol solutions at a rate of 5 °C/h. The large white crystals were then washed with diethyl ether and dried in the $N_2$ glovebox at 60 °C for 24 h.

**Doctor-blading of perovskite and poly-TPD films.** The $MAPbI_3$ were dissolved in DMF (Sigma Aldrich, 99.8% anhydrous) with concentration varying from 0.02 to 0.8 M. Then different molar amounts of FPMAI were added into $MAPbI_3$ precursors before blade-coating in air at 50 °C. The humidity during blade-coating was around 35%. The speed of applicator and $N_2$ knife was set as 150 and 50 mm/s, respectively. The pressure of $N_2$ knife was 0.2 MPa. The distance between the $N_2$ knife and substrate is 2 cm. After blade-coating, the perovskite films were transferred to $N_2$-filled glovebox immediately. For doctor-blading of poly-TPD layer, the poly-TPD solution (6 mg/ml in chlorobenzene) was doctor-bladed using the same method with perovskite film. The amount of poly-TPD and perovskite solutions were about 0.5 μl/cm$^2$.

**Spin-coating of perovskite films.** In all, 50% molar ratio excess FPMAI were added into $MAPbI_3$ solutions (0.2 M in DMF) before spin-coating. The spin coating rate was 4000 rpm, and toluene was dropped on the spinning substrate at around 6 s. About 300 μl of $MAPbI_3$ solution was used to prepare large-area (6 × 9 cm$^2$) films.

**Characterizations of perovskite films.** The thickness of perovskite films was measured using a Dektak XT profiler. The AFM measurements were conducted with tapping mode (Asylum Research MFP-3D-SA). The SEM measurements were conducted using field-emission SEM (Hitachi SU4800). XRD measurements were performed with a TTR-III (Rigaku) X-ray diffractometer with Bragg–Brentano parallel beam geometry, a diffracted beam monochromator, and a conventional Cu target X-ray tube set to 40 kV and 200 mA.

The PLQY of the perovskite films were measured using a Horiba Fluorolog-3 system with Petite Integrating Sphere. The excitation wavelength was 430 nm with an excitation intensity of 8 mW/cm$^2$. The TRPLs were taken using a Horiba time-correlated single-photon counting system. The samples were excited by a pulsed laser diode (NanoLED-Horiba) with a center wavelength of 369 nm, an excitation intensity of ~6 mW/cm$^2$, and a repetition rate that is less than the reciprocal of the measurement range. The time resolution was determined to be ~100 ps from the instrument response function. All decays were well fitted by 3 exponentials:

$$\text{decay}(t) = \sum_{n=1}^{3} B_n e^{-t/T_n}, \quad T_{\text{average}} = \frac{\sum_{n=1}^{3} B_n T_n^2}{\sum_{n=1}^{3} B_n T_n},$$ where $B_n$ are normalized coefficients

and $T_n$ are time constants. The microscale PL mapping was conducted on a PicoQuant MT100 FLIM System at room temperature. A 485-nm laser (PicoQuant LDH-P-C-405B) pulsed at 20 MHz with a power of 1.3 μW was coupled into the confocal microscope and focused onto the sample with a spot diameter of ~500 nm. The PL intensity was recorded by a hybrid PMT detector, and the PL mapping was conducted over a 10 × 10 μm region.

**PeLED fabrication and characterization.** Poly-TPD was dissolved in chlorobenzene at a concentration of 6 mg/ml. The Poly-TPD solution was spin-coated on ITO substrate at 1000 rpm, followed by annealing at 150 °C for 20 min. The poly-TPD layer was treated by $O_2$ plasma for 6 s to improve wetting. All blade-coated perovskite films are dried at 70 °C for 10 min to fully dry the film and ensure full reaction of the precursors without affecting the morphology and grain size

substantially. The TPBi, LiF, and Al layers were sequentially thermally evaporated on top of the perovskite film with thicknesses of 40, 1.2, and 100 nm, respectively. The device areas were 0.04, 1, and 28 cm$^2$, respectively.

The PeLEDs were measured in $N_2$ using a homemade motorized goniometer set-up consisting of a Keithley 2400 sourcemeter unit, a calibrated Si photodiode (FDS-100-CAL, Thorlabs), a picoammeter (4140B, Agilent), and a calibrated fiber optic spectrophotometer (UVN-SR, StellarNet Inc.). The distance between LED device and photodetector is 59.5 mm.

## Data availability
The data that support the findings of this study are available from the corresponding author upon reasonable request.

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

## Acknowledgements

We acknowledge the support from the National Natural Science Foundation of China (NSFC, Award # 51872274) and the Fundamental Research Funds for the Central Universities (WK2060190053). Z.X. thanks Barry P. Rand from Princeton University for his comment on the manuscript. We thank Guanyin Gao for the XRD measurements.

## Author contributions

Z.X. conceived the idea and supervised the project. S.C. fabricated and measured the PeLEDs. W.C. and Z.F. performed the thickness, AFM, SEM, PLQY, and TRPL characterizations. W.C synthesized MAI and FPMAI materials. X.X. and J.H. conducted the microscale PL mapping measurement. Y.L. and W.C. helped the fabrication and characterization of PeLEDs. J.C. calculated nucleus density in the films. Z.X. wrote the manuscript. All authors read and commented on the manuscript.

## Competing interests

The authors declare no competing interests.
