## [Peer Review File · Nature Communications]

REVIEWER COMMENTS

Reviewer #1 (Remarks to the Author):

the authors developed a method to prepare large-area perovskite LED devices based on a blade-coating method. the comparison between different parameters, characterizations has been elaborated nicely. I have seen this manuscript in its previous version, the improvement is noticeable. therefore would like to endorse my support for publication in Nature Communications.

Reviewer #2 (Remarks to the Author):

Title: Large-area, efficient perovskite light-emitting diodes via low-temperature blade-coating

Comments:

In this manuscript, the authors reported sol-gel engineering of low-temperature blade-coated methylammonium lead iodide (MAPbI₃) perovskite films, and show the precipitation, gelation, aging, and phase transformation stages with use of a diluted, organoammonium-excessed precursor for uniform films. Finally, they achieve a high external quantum efficiency of 16.1% and low material cost for the doctor-bladed PeLEDs. Although the work focuses on the engineering parts of doctor-bladed process, it shows an useful approach to fabricate large-area PeLEDs with high performances, which may be considered to publish after major revisions:

- (1) The authors adopt blade-coating process for the large perovskite film fabrication. The technological parameters are important to the film quality. In the manuscript, the used coating speed and temperature are simply stated as 150 mm/s and 50 °C respectively. The detail information about the optimized parameters is missing.
- (2) Since there is no antisolvent in the blade-coating process, the N₂ flow rate affects the solvent evaporation and film morphology a lot, the authors should better provide more information about the influences.
- (3) The organoammonium halide of 4-fluorophenylmethylammonium iodide (FPMAI) was employed to incorporate into the MAPbI₃ films. What is the specific role of FPMAI, as there are various organoammonium halides, such as phenylmethylammonium iodide, phenylethylammonium iodide, 1-naphthylmethylamine iodide, and so on.
- (4) In page 6. "As DMF fully evaporate out, the perovskite film with small grains forms." The boiling point of DMF is far higher than the blade-coating temperature of 50 °C. Is there an annealing process for the films to fully evaporate the DMF? It may be very difficult to fabricate the large uniform films under annealing treatment.
- (5) In the manuscript, the large films are prepared with an area of 6×9 cm², which is really larger than the reported ones. However, only a small volume of precursor (30 μL) is used for blade-coating. There is therefore a concern whether the films are directly covered on the poly-TPD or ITO substrates. Particularly, poly-TPD substrates is hydrophobic in nature.
- (6) The EL peaks blue shift with use of the dilute precursors. The authors shall explain the reasons such as quantum confinement effect, dielectric confinement effect, and etc. In addition, the authors attribute the high PeLED EQE with excess FPMAI to the small grains. How about the grain size?
- (7) In the manuscript, it states that "the devices without additive show relatively poor performances with EQE lower than 1%. This should be due to the low electron-hole binding energy of large MAPbI₃ grains." However, the exciton binding energies of MAPbI₃ is really small (≤25 meV, Nano Lett. 2015, 15, 3692–3696) even for the small grains. The authors shall elaborate the details.
- (8) The molar ratio of FPMAI is fixed at 50% for the highest EQE than other ratios (30%, 40%, Supplementary Fig. 9). It is noted that the EQE increases as increasing the ratios. How about the situation with molar ratio higher than 50%?

Response Letter

Reviewer #1

Comment: *“The authors developed a method to prepare large-area perovskite LED devices based on a blade-coating method. The comparison between different parameters, characterizations has been elaborated nicely. I have seen this manuscript in its previous version, the improvement is noticeable. Therefore, I would like to endorse my support for publication in Nature Communications.”*

Response: We thank the reviewer's time again, and appreciate the reviewer's acknowledgement of our work.

Reviewer #2

Comment: *“In this manuscript, the authors reported sol-gel engineering of low-temperature blade-coated methylammonium lead iodide (MAPbI₃) perovskite films, and show the precipitation, gelation, aging, and phase transformation stages with use of a diluted, organoammonium-excessed precursor for uniform films. Finally, they achieve a high external quantum efficiency of 16.1% and low material cost for the doctor-bladed PeLEDs. Although the work focuses on the engineering parts of the doctor-bladed process, it shows a useful approach to fabricate large-area PeLEDs with high performances, which may be considered to publish after major revisions”.*

Response: We thank the reviewer very much for your time to evaluate this work. We have addressed the comments point-by-point as follows.

Comment 1: *“The authors adopt blade-coating process for the large perovskite film fabrication. The technological parameters are important to the film quality. In the manuscript, the used coating speed and temperature are simply stated as 150 mm/s and 50 °C respectively. The detail information about the optimized parameters is missing.”*

Response: Thank the reviewer very much for pointing this out. We are sorry for missing the optimization data in the original manuscript, and have added all optimization details in the revised manuscript.

There are two regimes for the coating process, namely evaporation regime and Landau-Levich regime (Langmuir 2009, 25, 2554-2557, Nature Energy 2018, 3, 560-566). In the evaporation regime, the coating process is comparable to film drying timescale, and thus the perovskite starts to form slowly before the coating process is finished, resulting in rough films. In the Landau-Levich regime, the coating process is fast so that the film is still very wet after the coating process. In this case, the N₂ knife can be utilized to evaporate the DMF residue quickly out of the films, and uniform films can be formed.

We found that uniform films can be obtained using a relatively large range of coating speed, from 10 mm/s (lowest limit of our blade-coating equipment) to somewhere around 150 mm/s, both of which are in the Landau-Levich regime according to our observation. It should be noted that, in

the Landau Levich regime, the thickness of the wet film is positively related to the coating speed. The coated film became very thin, looks almost transparent, using 10 mm/s coating speed. When we increased the coating speed to 250 mm/s, the N₂ knife drove the wet layer flowing on the substrate, resulting in non-uniform films. Therefore, we select a relatively high coating speed of 150 mm/s in this work, which is preferred for large-area mass production.

Supplementary Fig. 1 | **a**, Schematics of the blade-coating setup. **b**, Statistics of film thickness made from 0.04 M solution and coating speed of 150 mm/s. **c-d**, Photo images of blade-coated films with coating speed of 10 mm/s (c), 50 mm/s (d), and 250 mm/s (e). A N₂ knife was used for all films. The lower coating speed (e.g. 10 mm/s) can also result in uniform but very thin films, while higher coating speed of 250 mm/s results in non-uniform films due to the flowing of solution on the substrate. The substrate sizes are 6 cm × 9 cm.

As for the coating temperature, the crystallization rate is slow at room temperature even with the help of N₂ knife, resulting in rough films with larger grain sizes. On the contrary, if the coating temperature is too high (e.g. 80 °C), the film starts to dry before we applied the N₂ knife (Landau Levich regime). And therefore, the film also became very rough. According to our observation, somewhere around 50 °C is the optimized temperature to form uniform films.

Supplementary Fig. 2 | SEM images of perovskite films blade-coated at different temperatures. The coating speed for all films is 150 mm/s. The scale bars are 100 nm.

We modified the revised manuscript accordingly:

1. We added the photo images for different coating speed in Supplementary Fig. 1, and added the SEM images for different coating temperature as Supplementary Fig. 2 in the revised SI.
2. In the page 3, line 19 of the revised manuscript, we added “*It should be noted that the blade-coating speed, temperature, and the N₂ knife pressure affect the film morphology dramatically (Supplementary Fig. 1-3). An ultra-fast coating speed of 150 mm/s, corresponding to Landau-Levich mode that the as-coated layer is still wet right after blading²⁹, a low temperature of 50 °C, and a N₂ knife pressure of 0.2 MPa are adopted for the coating process. Details of the optimization process can be found in the Supplementary Information.*”

Comment 2: “*Since there is no antisolvent in the blade-coating process, the N₂ flow rate affects the solvent evaporation and film morphology a lot, the authors should better provide more information about the influences.*”

Response: Yes, the N₂ flow rate affects the solvent evaporation speed and thus the film morphology. As shown in the figures below, the film became very hazy if N₂ knife pressure reduced from 0.2 MPa to 0.1 MPa, resulted from the slower evaporation rate of DMF solvent. SEM images show that the films are not uniform with many pinholes. When the pressure of N₂ knife was increased to 0.4 MPa, the N₂ knife drove perovskite wet film flowing on the substrate, resulting in a non-uniform film. Therefore, we used 0.2 MPa for the blade-coating process.

Supplementary Fig. 3 | **a,b**, A photo image of a blade-coated film fabricated using a N₂ knife pressure of 0.1 MPa (**a**), and SEM images taken at different positions (**b**). The scale bars are 2 μm. **c,d**, A photo image (**c**) and optical microscope images at different positions (**d**) of a perovskite film fabricated using a N₂ knife pressure of 0.4 MPa. The scale bars are 0.4 mm.

We modified the revised manuscript accordingly:

1. We added the influence of N₂ knife pressure data as Supplementary Fig. 3 in the revised SI.
2. In the page 3, line 19 of the revised manuscript, we added “*It should be noted that the blade-coating speed, temperature, and the N₂ knife pressure affect the film morphology dramatically (Supplementary Fig. 1-3). An ultra-fast coating speed of 150 mm/s, corresponding to Landau-Levich mode that the as-coated layer is still wet right after blading²⁹, a low temperature of 50 °C, and a N₂ knife pressure of 0.2 MPa are adopted for the coating process. Details of the optimization process can be found in the Supplementary Information.*”

Comment 3: “*The organoammonium halide of 4-fluorophenylmethylammonium iodide (FPMAI) was employed to incorporate into the MAPbI₃ films. What is the specific role of FPMAI, as there are various organoammonium halides, such as phenylmethylammonium iodide, phenylethylammonium iodide, 1-naphthylmethylamine iodide, and so on.*”

Response: This is a good point. We have investigated the effect of the chemical structure of bulky organoammonium halides on the optoelectronic properties of perovskite films and the performance of MAPbI₃ based PeLEDs in our previous works (Xiao, Z. *et al. Adv. Funct. Mater.* 29, 1807284, 2019). We compared bulky organoammonium ligands with an alkyl chain, specifically butylammonium (BA), with that of benzylammonium where the phenyl ring has 0, 1, and 2 fluorine

substitutions. As shown in **Figure I**, the films with phenyl ring ligands have higher PLQY than the film with the BAI additive which may be due to their better capping effect. The film with 4-F-PMAI has the highest PLQY. There are two possible reasons. Firstly, the 4-F-PMAI cations have a better tendency in packing face-on than other allotropes with different F positions. This may enhance the crystallinity of the perovskite nanocrystal films and reduce trap density. Secondly, the electron-withdrawing fluoro group can increase the polarity of additives and its bonding with Pb-I octahedra, and thus the stability of the perovskite films. Therefore, we use 4-F-PMAI directly in this work.

[REDACTED]

Figure I. **a**, Molecular structure of the bulky ligands. **b,c**, PLQY as a function of excitation intensity of the MAPbI₃ films (**b**) and EQE versus current density of PeLEDs (**c**) with different bulky organoammonium halides as a surfactant. *Copied from Adv. Funct. Mater.* 29, 1807284, 2019.

Comment 4: “In page 6. “As DMF fully evaporate out, the perovskite film with small grains forms.” The boiling point of DMF is far higher than the blade-coating temperature of 50 oC. Is there an annealing process for the films to fully evaporate the DMF? It may be very difficult to fabricate the large uniform films under annealing treatment.”

Response: Yes, there was an annealing process (70 °C for 10 min) for both large-area and small-area films to fully evaporate DMF residue and improve the film crystallinity. A common hot plate with small temperature fluctuations was used for the annealing process, as shown in Figure II below.

Figure II. a, A photo image of the hot plate used for annealing of perovskite films. **b,** Temperature mapping of the hotplate set at 70 °C. The hot plate surface was divided by 2 cm×2 cm small areas and the temperature of each small area was measured separately. The figure is plotted by the 2D interpolation method.

Comment 5: “*In the manuscript, the large films are prepared with an area of 6×9 cm², which is really larger than the reported ones. However, only a small volume of precursor (30 μL) is used for blade-coating. There is therefore a concern whether the films are directly covered on the poly-TPD or ITO substrates. Particularly, poly-TPD substrates is hydrophobic in nature.*”

Response: Yes, the wettability of the poly-TPD is critical for the film morphology. The poly-TPD films were treated by O₂ plasma for 6 s to improve the wettability. We could not get good films on non-wetting poly-TPD films without O₂ plasma treatment, as shown in Figure III below.

Figure III. A photo image of the blade-coated perovskite film on a fresh poly-TPD film without O₂ plasma treatment.

Comment 6: “*The EL peaks blue shift with use of the dilute precursors. The authors shall explain the reasons such as quantum confinement effect, dielectric confinement effect, and etc. In addition, the authors attribute the high PeLED EQE with excess FPMAI to the small grains. How about the grain size?*”

Response: Thank the reviewer very much for this valuable comment. We followed the reviewer’s comments and performed high-resolution SEM measurement on the optimized film (0.04 M with 50% FPMAI). As shown below, the grain sizes are around 10 nm. Considering the Bohr radius of MAPbI₃ perovskite is around 2.2 nm (*Solid State Commun.* 127, 619-623, 2003), we think the EL peaks blue shift is dominated by dielectric confinement effect by the bulky organoammonium ligands at the perovskite nanocrystal surface and/or adjacent organic electron/hole transporting layers, rather than quantum confinement effect.

Supplementary Fig. 6 | A SEM top morphology image of the optimized perovskite film.

We modified the revised manuscript accordingly:

1. We added the SEM image as Supplementary Fig. 6 in the revised SI.
2. In page 4, line 23 of the revised manuscript, we added “A high-resolution SEM image shows that the grain sizes are around 10 nm (Supplementary Fig. 6), comparable to the optimized spin-coated films²⁰.”
3. In page 9, line 5 of the revised manuscript, we added “The blue shift of EL peaks for the ultra-thin perovskite films using dilute precursors should result from dielectric confinement effect by the bulky ligands self-assembled at the perovskite nanocrystal surface and/or adjacent organic electron/hole transporting layers³³, considering the grains sizes are larger than the Bohr radius³⁴.”

Comment 7: “In the manuscript, it states that “the devices without additive show relatively poor performances with EQE lower than 1%. This should be due to the low electron-hole binding energy of large MAPbI₃ grains.” However, the exciton binding energies of MAPbI₃ is really small (≤ 25 meV, *Nano Lett.* 2015, 15, 3692-3696) even for the small grains. The authors shall elaborate the details.”

Response: There are large fluctuations in the reported values of the exciton binding energy of MAPbI₃ films, ranging from 19 to 62 meV (*Adv. Energy Mater.* 2020, 10, 1903659). We measured the exciton binding energy of our films using temperature-dependent PL. The PL intensities were fitted using an Arrhenius formula $I(T) = I_0 / (1 + A_0 \exp(-E_b / (k_b T)))$, where E_b is exciton binding energy, A_0 is a constant, I_0 is the intensity at 0 K, and k_b is the Boltzmann constant. As shown in the following figure, the E_b increases from 37.7 meV to 50.0 meV after incorporation of FPMAl.

Figure IV. Temperature-dependent integrated PL intensity of the perovskite film W/O FPMAI (a), and with FPMAI (b). The solid lines are fitted based on the Arrhenius equation.

Comment 8: “The molar ratio of FPMAI is fixed at 50% for the highest EQE than other ratios (30%, 40%, Supplementary Fig. 9). It is noted that the EQE increases as increasing the ratios. How about the situation with molar ratio higher than 50%?”

Response: Thank the reviewer for this comment. We provided data of the PeLEDs with 60% FPMAI in the revised manuscript. Generally, the EQE of the device with too much FPMAI will decrease due to the formation of too much low-dimensional perovskite phases because of their strong electron-phonon coupling (*Nat. Photon.*, 11, 108, 2017).

Supplementary Fig. 12 | Optimization of the FPMAI molar ratios. a,b, J-V (a) and EQE (b) curves of the PeLEDs with different molar ratios of FPMAI.

We modified the revised manuscript accordingly:

1. We updated the Supplementary Fig. 12 in the revised SI (Supplementary Fig. 9 of the original manuscript).

REVIEWERS' COMMENTS

Reviewer #2 (Remarks to the Author):

In the revised manuscript, the authors have carefully addressed the reviewer concerns. They demonstrate efficient perovskite light-emitting diodes based on the strategy of low-temperature blade-coating. Thus, it can be published in Nature Communications as is.

Response Letter

Reviewer #2

Comment: *“In the revised manuscript, the authors have carefully addressed the reviewer concerns. They demonstrate efficient perovskite light-emitting diodes based on the strategy of low-temperature blade-coating. Thus, it can be published in Nature Communications as is.”*

Response: We thank the reviewer’s time again, and appreciate the reviewer’s acknowledgement of our work.